# Embodied Contrastive Learning with Geometric Consistency and Behavioral Awareness for Object Navigation

## ABSTRACT

Object Navigation (ObjcetNav), which enables an agent to seek any instance of an object category specified by a semantic label, has shown great advances. However, current agents are built upon occlusion-prone visual observations or compressed 2D semantic maps, which hinder their embodied perception of 3D scene geometry and easily lead to ambiguous object localization and blind exploration. To address these limitations, we present an Embodied Contrastive Learning (ECL) method with Geometric Consistency (GC) and Behavioral Awareness (BA), which motivates agents to actively encode 3D scene layouts and semantic cues. Driven by our embodied exploration strategy, BA is modeled by predicting navigational actions based on multi-frame visual images, as behaviors that cause differences between adjacent visual sensations are crucial for learning correlations among continuous visions. The GC is modeled as the alignment of behavior-aware visual stimulus with 3D semantic shapes by employing unsupervised contrastive learning. The aligned behavior-aware visual features and geometric invariance priors are injected into a modular ObjectNav framework to enhance object recognition and exploration capabilities. As expected, our ECL method performs well on object detection and instance segmentation tasks. Our ObjectNav strategy outperforms state-of-the-art methods on MP3D and Gibson datasets, showing the potential of our ECL in embodied navigation. The experimental code is available as supplementary material.

## CCS CONCEPTS

• **Computing methodologies** → **Computer vision tasks**; **Knowledge representation and reasoning**; **Vision for robotics**.

## KEYWORDS

Object Navigation, Contrastive Representation Learning, Geometric Consistency, Behavioral Awareness, Embodied AI

**ACM Reference Format:**
Anonymous Author(s). 2024. Embodied Contrastive Learning with Geometric Consistency and Behavioral Awareness for Object Navigation. In *Proceedings of the 32th ACM International Conference on Multimedia (MM '24), October 28–November 1, 2024, Australia, Melbourne.* ACM, New York, NY, USA, 15 pages. https://doi.org/XXXXXXX.XXXXXXX

## 1 INTRODUCTION

Object Navigation (ObjectNav) task [3, 23] requires an agent to navigate through a previously unknown 3D scenario to find an object instance, according to a semantic label. Existing work has made great advances in visual representations [18, 19, 63, 67], data augmentation techniques [40, 46], and auxiliary tasks [33, 58] for pre-training. Their core ideas are fully exploiting scene layouts and semantic contexts to enhance agents' object localization or scene exploration capabilities. Some methods [18, 19, 41, 65] speculate on correlations among historical visual features for ObjectNav decision-making by emphasizing spatio-temporal awareness of visual observations. Although promising progress has been made, domestic scenes are characterized by substantial occlusion, which poses challenges for agents to accurately localize object goals and efficiently explore scenarios. Moreover, agents typically establish high-level awareness of objects by moving around and perceiving them from different angles and distances. For instance, learning about basic physical concepts for object localization, such as large and long, requires moving beyond image-based observations.

Research in behavioral psychology [42, 51] has shown that many animals maintain spatial representations of their environments while navigating. Inspired by this, some other methods attempt to develop Topological Scene Representations (TSRs) [16, 17, 32, 34, 60, 63] or 2D contextual semantic maps [9, 22, 23, 45, 61] based on visual images to balance exploration and exploitation better, as shown in Fig. 1 (a). Nodes in TSRs typically consist of abstract visual or object features [16, 34]. Edges in TSRs usually involve discrete semantic or geometric relationships (e.g., the pillows are in the bed, and the mouse is used to operate the computer) [32, 60]. However, the abundant geometric and semantic relations among objects should be a large relational space and thus difficult to model with discrete TSRs exhaustively. The 2D contextual semantic maps somewhat reconstruct the layouts and semantic patterns of the scenarios and can provide agents with compressed materials to formulate the continuous relational space [11]. However, RGB image-based semantic segmentation errors may lead to low-quality and ambiguous 2D semantic maps, which severely impair the ObjectNav performance, please see Section A of the supplementary material for more details.

To alleviate the above problems, we propose an Embodied Contrastive Learning (ECL) method with Geometric Consistency (GC) and Behavioral Awareness (BA) to motivate agents to actively explore 3D scene layouts and encode semantic cues. Instead of modeling the correlations among visual features from a spatio-temporal aware perspective, we advocate inferring the relevance among visual features at the root by predicting intermediate actions from consecutive visual frames. We believe that the behaviors that lead to differences between two adjacent observations are crucial for learning the relations between two visions, as shown in Fig. 1 (b). Our BA modeling is more concise than predicting visual features from action sequences since the observation space consisting of

 

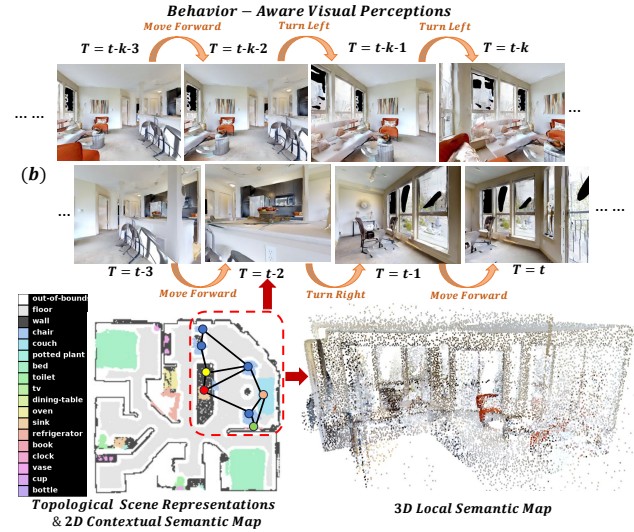

**Figure 1: (a) An illustration of a 2D contextual semantic map and a local TSR. (b) An illustration of behavior-aware visual perceptions. The changes in visual sensations are caused by navigational behaviors. (c) An illustration of 3D local semantic maps used for geometric-aware contrastive representation learning, corresponding to the local region in (a).**

high-dimensional sensor inputs tends to be large and variable, while the action space is small, discrete, and relatively fixed.

When searching for a specific object, humans usually take sequential actions to actively transform their Field of View (FoV), locating the target and exploring the scene by aligning their visual senses with 3D space. Inspired by this, we empower agents to reconstruct local 3D spatial structures and semantic patterns during navigation, as shown in Fig. 1 (c). The 3D local scene priors allow agents to learn rich scene representations by immersively aligning the visual features encoded by the behavior-aware visual encoder with the 3D scene features encoded by the 3D PointCloud (PCL) encoder. As a result, the 2D scene understanding is enhanced by introducing geometric and view-invariant priors into the behavior-aware 2D visual features. In a nutshell, the GC is modeled as the alignment of behavior-aware visual perceptions with 3D semantic shapes by employing unsupervised contrastive learning.

Notably, to adequately mimic the situational interactions of humans with 3D scenes, the above BA and GC-based contrastive representation learning is performed in an embodied manner. In particular, we propose a curiosity and action-aware exploration policy $E^2$-CL, which continually motivates agents to adopt diverse actions to discover novel visual perceptions. On the one hand, the semantic-rich visual stimulus facilitates agents to carry out more comprehensive and robust 2D-3D scene representation learning. On the other hand, the diverse action-vision data pairs collected online provide rich learning materials and feature bases for BA modeling. With the collection of novel and complicated action-vision pairs, the visual frames-based action prediction will be more challenging. Therefore, the BA modeling will be gradually enhanced in this adversarial learning process.

During the experimental phase, we first validate the superiority of our ECL on generic object detection and instance segmentation

tasks. Particularly, the visual encoder pre-trained by ECL is further retrained to solve these two tasks. The great performance of our approach on both tasks reflects ECL's expertise in object recognition, which will be further migrated to the ObjectNav task. In addition, the pre-trained visual and PCL encoders are integrated into a modular ObjectNav strategy, which is compared with state-of-the-art (SOTA) ObjectNav methods on Matterport3D (MP3D) [7] and Gibson [54] datasets. Concretely, our method improves the ObjectNav success rate by 1.4% ∼ 6.2% and 0.8% ∼ 3.1% on the two datasets, respectively. Sufficient ablation studies demonstrate the substantial contributions of the individual components in our method. Overall, the contributions of this paper are as follows:

**(1)** An embodied contrastive representation learning method with BA and GC is proposed. The BA modeling helps agents take informed navigational actions based on behavior-aware visual perceptions. The GC modeling infuses agents with 3D geometric invariance priors. **(2)** A curiosity and action-aware exploration strategy is proposed to support embodied ECL. The diverse action-vision pairs collected online provide rich feature bases for the BA and GC modeling. **(3)** Sufficient comparative and ablative studies on object detection, instance segmentation, and ObjectNav tasks demonstrate the superiority of our ECL and $E^2$-CL methods.

## 2 RELATED WORK

**(1) Spatio-Temporal Visual Modeling for Visual Navigation**. Human beings can naturally navigate in new environments, which requires us to find parallels between the new observations and our past experiences. Inspired by this, visual modeling of spatio-temporal awareness [13, 18, 19, 30, 36, 41, 65, 66] can provide agents with historical contextual information for navigation. Its core concept is to implicitly encode the visual semantic clues, the relative spatial information among objects, and the temporal correlations among multiple visual frames, using recurrent neural networks [18, 19, 41] or Transformers [13, 30, 36, 65, 66]. Some improved methods [41] exploit spatio-temporal attention mechanisms to filter keyframes and intelligently focus on semantic and spatial cues that are most relevant to the navigation goal. One of our main insights is that variations in visual perceptions are the consequence of active navigational behaviors. Therefore, unlike spatio-temporal visual modeling that extrapolates dynamic correlations back from observations, we believe the modeling of behavioral patterns can help characterize those correlations at the source.

**(2) TSRs and Semantic Maps for Visual Navigation.** TSRs have continuously shown improvements in various visual semantic tasks. Visual language navigation [14, 21, 35, 52] typically embeds panoramic visual features into the topological graph's nodes and represents the topological graph's edges as combinations of relative positions and orientations between the nodes. Although the topology retains the partial geometry of the scene, the relationships and dynamic transitions between nodes should be encoded with richer geometric and semantic cues. ObjectNav [18, 19, 64] usually injects object features detected from visual perceptions into the topological graph's nodes. The difference is that the topological graph's edges include not only geometric relative positions but also semantic relationships, e.g., a mouse is used to operate a computer. Nevertheless, some studies of continuous scene representations

[11, 20] have shown that such discrete relational modeling is prone to inductive bias in spatial and semantic understanding.

Since semantic maps preserve fine-grained scene layouts and semantic patterns, they can alleviate the inductive bias by providing navigation agents with richer scene representations. By projecting high-dimensional features encoded from a neural network to a top-down map, existing works [6, 26] try to generate a deep feature map, which is used for reconstructing scenes, predicting semantic maps, and navigating. The semantic map is a type of occupancy map [44], which indicates whether a point is occupied, represents the semantic categories of objects, and provides location information for robotic navigation and exploration. By decoupling ObjectNav tasks into object localization and scene exploration subtasks, a series of 2D semantic map-based modular methods [39, 45, 61, 62] have been proposed and achieved promising performance. Although 2D contextual semantic maps somewhat reconstruct the layouts and semantic patterns of the scenarios, they lose the 3D geometric structure that is critically important for embodied navigation.

Although existing work achieves impressive performance on visual navigation by introducing 3D semantic maps [8, 62] and bird's-eye views [1, 38], how to fully mine and exploit 3D geometric features is still a challenging and open topic. To release the agents from the 2D observation space, this paper proposes a novel 2D-3D contrastive representation learning method with geometric awareness by aligning rich visual features with 3D scene priors in an embodied manner.

**(3) Unimodal and Multimodal Representation Learning for Visual Navigation.** Unimodal representation learning has emerged with significant success in visual navigation tasks. OVRL [55] employs the concept of knowledge distillation [5] to learn navigation-friendly visual representations from pure visual images in an offline manner. In contrast, CRL [20] presents an embodied adversarial contrastive learning technique to encourage agents to actively explore novel surroundings for learning robust visual representations online. To alleviate the typically substantial occlusions in visual images, Chen et al. [11] employed offline contrastive learning to extract continuous relations among objects from 2D semantic maps. In terms of TSRs, a novel scene graph contrastive loss [50] is proposed to encourage representations that encode objects' semantics, relationships, and history. Unlike the above methods, one of our main insights is to merge the merits of different modalities to achieve richer and more robust scene representations.

Multimodal representation learning gains attraction due to its ability to share modality-specific contexts. Humans naturally build spatially meaningful cognitive maps in their brains during navigation. Inspired by this, $Ego^2$-Map [27] proposes a navigation-specific method for learning visual representations by aligning egocentric views with 2D semantic maps in a cross-modal manner. Although this idea is commendable, ObjectNav agents are born and work in 3D scenes. In this work, we experimentally prove that 3D geometric awareness is crucial for ObjectNav decision-making by comparing our method with $Ego^2$-Map. Inspired by recent work on 2D-3D multimodal representation learning [2, 12, 29, 57], this paper proposes a GC-based ECL technique that encourages agents to actively exploit 3D geometric and semantic priors. Our inspiration comes from Pri3D [29], which introduces a contrastive learning method for multi-view RGB frames and 3D scene scans. Unlike Pri3D, our ECL

method can organically fuse the geometric features in 3D semantic shapes with behavior-aware visual features. The features from both modalities will be utilized to enhance ObjectNav. In addition, our ECL is conducted online in an embodied fashion, with the aim of active learning through interactions with the scenarios.

## 3 METHODOLOGY

### 3.1 Problem Statement and Overview

**Problem Statement.** In an unknown environment, the ObjectNav task requires the agent to navigate to an instance of the specified target category. As initialization, the agent is located randomly without access to a pre-built environment map and is given a target category $c_{target} \in \{1, 2, ..., C\}$, where $C$ is the number of possible target categories. For each navigation state $s_t$, the agent receives noiseless onboard sensor readings, including egocentric RGB-D images $\{o_t, d_t\}$ and a 3-DoF pose $\{x, y, \theta\}$ (2D position and 1D orientation) relative to the starting of the episode. Then the agent predicts its action $a_t$ for movement in a discrete action space, consisting of $Move\_Forward$, $Turn\_Left$, $Turn\_Right$, and $Stop$. An episode is considered successful if the agent executes $Stop$ within 1.0 m of a target object and the object can be viewed from the agent's position. Each episode has a time limit of 500 steps.

**Overview.** Fig. 2 and Fig. 3 provide overviews of the proposed ECL method and our modular ObjectNav strategy, respectively. A curious contrastive reward $\mathcal{R}_{Exp}$ with action awareness is designed for the embodied exploration in ECL, which motivates agents to actively explore the scene and consistently gather novel visual images (§3.2). We believe navigation behavior is one of the main factors affecting embodied agents' visual perceptions, the BA is modeled by using continuous visual frames to predict the intermediate actions (§3.3). Meanwhile, a PCL-based semantic mapping method is employed to project the semantically segmented RGB images to a 3D semantic map based on the depth images, the camera parameters, and the agent's poses. This paper focuses on whether 3D scene priors can enhance visual semantic navigation, thus a contrastive loss $\mathcal{L}_{ECL}$ is proposed for GC modeling by aligning behavior-aware visual features with corresponding 3D scene priors (§3.3). Finally, the visual encoder $F_\theta$ and PCL encoder $F_\varphi$ pre-trained by ECL are migrated to downstream tasks to boost the performance of object detection, instance segmentation, and ObjectNav (§3.4).

In a modular fashion, we decompose the ObjectNav pipeline into three phases: BA and GC-based scene representation, prediction-based high-level goal selection, and deterministic low-level planning, as shown in Fig. 3. Following existing modular approaches [9, 23, 45, 61, 62], our ObjectNav agent uses a top-down 2D semantic map as its internal environmental representation. The ObjectNav task is decoupled into two sub-tasks: object localization and scene exploration.

### 3.2 Embodied Exploration with Contrastive Learning ($E^2$-CL)

This work advocates that agents learn to build environmental cognition by continuously interacting with their surroundings as humans do. To ensure adequate interactions with diverse scenarios, a curiosity-driven exploration policy $\pi_\theta$ named $E^2$-CL is designed to motivate embodied agents to actively explore the scenarios. $\pi_\theta$

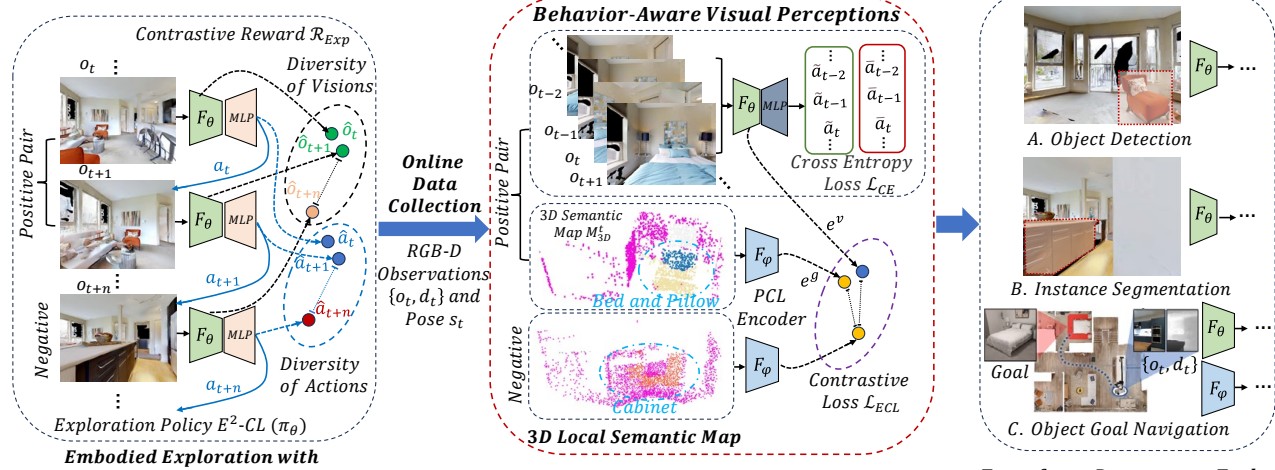

**Figure 2: (Left: §3.2)** Driven by the exploration policy $\pi_\theta$, our embodied agent actively seeks novel visual observations by adopting diverse actions to maximize the curious contrastive reward $\mathcal{R}_{Exp}$. **(Middle: §3.3)** Our ECL method aligns the behavior-aware visual features encoded by the visual encoder $F_\theta$ with the 3D features encoded by the PCL encoder $F_\varphi$ to model GC by minimizing the contrastive loss $\mathcal{L}_{ECL}$. The BA is modeled by minimizing the cross-entropy loss $\mathcal{L}_{CE}$ between the predicted actions and the real actions. **(Right: §3.4)** The pre-trained $F_\theta$ and $F_\varphi$ are transferred to downstream tasks for retraining and task-specific performance evaluation.

drives agents to consistently discover novel visual perceptions and facilitates the learning of semantically enriched scene representations. Specifically, the exploration policy $\pi_\theta$ consists of a visual encoder $F_\theta$ and a Multilayer Perceptron (MLP) projection head used for predicting exploration actions, as shown in Fig. 2 (Left). For the RGB visual observations $O = \{o_k\}_{k=1,2,...,N}$ collected by the agent during exploration, different image augmentation methods are applied to each image to obtain pairs of augmented images $\hat{O} = \{\hat{o}_k^1, \hat{o}_k^2\}_{k=1,2,...,N}$. $N$ denotes the number of collected RGB images. Each augmented image is encoded using $F_\theta$ and followed by L2 normalization, which is formulated as $\hat{z}_k^* = Normalize(F_\theta(\hat{o}_k^*))$. To maximize the diversity of visions, the following reward signal is used to train $\pi_\theta$ by maximizing the similarity between the augmented images in the same pair and minimizing the similarity between the augmented images in different pairs:

$$\mathcal{R}_V = \frac{1}{N} \sum_{k=1}^{N} log \frac{exp(sim(\hat{z}_k^1, \hat{z}_k^2)/\tau)}{\sum_{z^- \in \hat{Z}\setminus\{\hat{z}_k^1\}} exp(sim(\hat{z}_k^1, z^-)/\tau))}, \quad (1)$$

where $\hat{Z} = \{\hat{z}_k^1\}_{k=1,2,...,N}$, $\tau$ is a softmax temperature scaling parameter, and $sim(\cdot, \cdot)$ corresponds to the dot product.

In practice, we find the agent may only slyly perform steering actions in place to greedily maximize the reward described in Equation (1). To alleviate this problem, another reward signal is designed to maximize the diversity of actions:

$$\mathcal{R}_\mathcal{A} = \frac{1}{N} \sum_{k=1}^{N} \sum_{a^- \in \hat{\mathcal{A}}\setminus\{\hat{a}_k\}} sim(\hat{a}_k, a^-), \quad (2)$$

where $\hat{\mathcal{A}} = \{\hat{a}_k\}_{k=1,2,...,N}$, $\hat{a}_k = MLP(a_k)$, and $\mathcal{A} = \{a_k\}_{k=1,2,...,N}$ denotes the actions taken to collect visual observations. Overall, the curiosity and action awareness-driven reward $\mathcal{R}_{Exp} = \mathcal{R}_V + \alpha\mathcal{R}_A$ is employed as the total reward signal for our $E^2$-CL. $\alpha$ is a weight

used to balance the two rewards. By doing so, the agent is motivated to discover diverse RGB images by taking diverse actions.

Notably, the exploration policy is trained to maximize the following cumulative reward by utilizing the Proximal Policy Optimization (PPO) [48]:

$$\max_\theta \mathbb{E}_{x \sim \pi_\theta} \left[ \sum_{t=0}^{T} \mathcal{R}_{Exp}(F_\theta, x) \right]. \quad (3)$$

In particular, $\pi_\theta$ is trained by optimizing the objective $L(\theta) = \mathbb{E}[min(c_t(\theta)A_t, clip(c_t(\theta), 1-\epsilon, 1+\epsilon)A_t)]$, where the clip ratio $c_t = \frac{\pi_\theta(a_t|s_t)}{\pi_{\theta_{old}}(a_t|s_t)}$ and the advange $A_t$ is computed utilizing the value function $V(s_t)$. During exploration, $\pi_\theta$ is optimized by using the collected minibatches of data from PPO. Please see Section B of the supplementary material for more details. As shown in Fig. 2, the RGB-D images $\{o_t, d_t\}$ and agent's poses $s_t$ collected by $E^2$-CL are used for the subsequent online ECL.

## 3.3 ECL with Geometric Consistency and Behavioral Awareness

**Behavioral Awareness Modeling.** When searching for a specific object (e.g., a key), humans usually actively transform their FoVs or move forward to localize the object instance. Inspired by this, we propose to model the correlations between behaviors and visions (the long-horizon dynamic transitions between visual frames) by predicting action sequences based on successive multi-frame visual perceptions, as shown in Fig. 2. Notably, our exploration strategy $E^2$-CL tends to adopt diverse actions to discover novel visual stimuli, which provides rich behavior-vision data for BA modeling. To be more specific, a neural network consisting of the visual encoder $F_\theta$ and an MLP projection head is utilized to predict $l$ intermediate navigation actions $\{\tilde{a}_i\}_{i=t}^{t+l-1}$ from $l+1$ RGB visual frames $\{o_t\}_{i=t}^{t+l}$. This procedure is accompanied by the embodied exploration so that

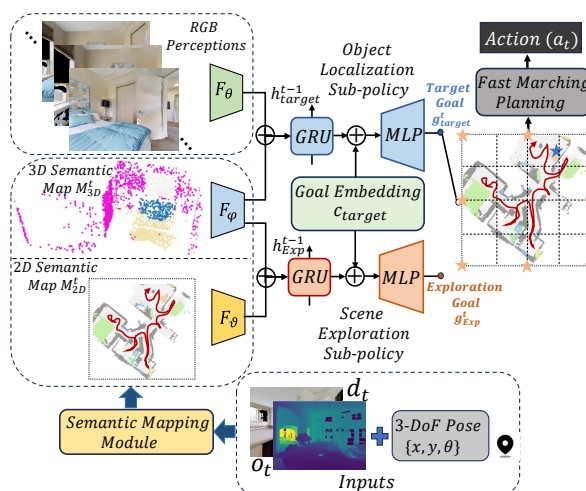

**Figure 3: The ObjectNav strategy takes RGB-D images, agent's pose, and object goal category as inputs. A PCL-based semantic mapping module is employed to build a 3D semantic map $M_{3D}^t$ along with a projected 2D semantic map $M_{2D}^t$ based on semantically segmented RGB-D images and poses. The pretrained visual encoder $F_\theta$ and PCL encoder $F_\varphi$ are utilized for object localization. $F_\varphi$ and another 2D map encoder $F_\vartheta$ are used for scene exploration. A deterministic local planning policy is utilized to drive the agent to the target goal $g_{target}^t$ or exploration goal $g_{Exp}^t$.**

the agent's real navigation actions $\{\bar{a}_i\}_{i=t}^{t+l-1}$ can be collected as supervisory signals. The cross-entropy loss $\mathcal{L}_{CE}$ is used to optimize the neural network:

$$\mathcal{L}_{CE} = -\frac{1}{l}\sum_{i=0}^{l-1}\bar{a}_i log\tilde{a}_i. \tag{4}$$

It is worth noting that the agent's action space is small and discrete while the observation space is relatively large and variable. Therefore, our BA modeling is concise and manageable in complexity compared to the modeling technique of predicting visual observations from navigation actions.

**Geometric Consistency Modeling.** To align the 2D behavior-aware visual features with the 3D scene priors, the semantically segmented RGB images are projected into a 3D semantic map in the form of PCL based on the camera's parameters and the corresponding depth images. At time step $t$, the past $l + 1$ frames of visual observations $\{o_i, d_i\}_{i=t-l}^t$ and poses $\{s_i\}_{i=t-l}^t$ are utilized to construct a 3D local semantic map $M_{3D}^t \leftarrow \{(P_p^t, P_s^t)\} \in \mathbb{R}^{Q^t \times (C+3)}$ for the consistency of 2D-3D features. Here, $P_p^t \in \mathbb{R}^3$ and $P_s^t \in \mathbb{R}^C$ denote each point's position and semantic category, respectively. $C$ and $Q$ denote the number of semantic categories and the number of PCL in $M_{3D}^t$, respectively. In practice, the off-the-shelf models [24, 31] are employed to obtain semantic segmentations from $\{o_i\}_{i=t-l}^t$ and combine the depths $\{d_i\}_{i=t-l}^t$ to generate 3D semantic PCL. However, each semantic point may probabilistically belong to multiple different semantic categories. Therefore, we suggest employing a max-fusion mechanism [62] to merge the temporal-variant semantic PCL to ensure consistent scene semantics.

Humans often efficiently localize specific object targets in an embodied manner by actively matching their visual senses with

3D scene structures. Inspired by this, a multimodal contrastive loss $\mathcal{L}_{ECL}$ is proposed to push the 2D and 3D features describing the same spatial patterns closer to each other while pushing the 2D and 3D features describing different spatial patterns farther away from each other. More specifically, $\mathcal{L}_{ECL}$ aligns the behavior-aware visual features $e^v$ with the 3D geometric-aware features $e^g$ by constructing a common 2D-3D space for the visual encoder $F_\theta$ and the PCL encoder $F_\varphi$, as shown in Fig. 2. To this end, the multimodal contrastive learning loss is as follows:

$$\mathcal{L}_{ECL} = \mathcal{L}_{cross}(e^v, e^g) + \mathcal{L}_{cross}(e^g, e^v), \tag{5}$$

$$\mathcal{L}_{cross}(e^v, e^g) = \frac{1}{2N}\sum_{i=1}^N -log\frac{exp(sim(e_i^v, e_i^g)/\tau)}{\sum_{e^- \in e^g \backslash \{e_i^g\}} exp(sim(e_i^v, e^-)/\tau)}. \tag{6}$$

$\mathcal{L}_{ECL}$ can not only transfer geometric information from 3D to behavior-aware 2D features but also transfer semantic details from 2D to 3D features. Overall, the visual encoder $F_\theta$ and the PCL encoder $F_\varphi$ are trained by using the loss $\mathcal{L}_V = \mathcal{L}_{CE} + \beta\mathcal{L}_{ECL}$ and the loss $\mathcal{L}_{ECL}$, respectively. $\beta$ is a weight used to balance the two losses. Moreover, $F_\theta$ is also trained by utilizing PPO as described in subsection 3.2.

## 3.4 Transfer to Downstream Tasks

**Object Detection and Instance Segmentation.** Before transferring the pre-trained visual encoder $F_\theta$ to the ObjectNav task, $F_\theta$ is employed to solve the object detection and instance segmentation tasks to verify its expertise in terms of object recognition. During ECL, the semantically rich and novel visual samples collected by $E^2$-CL are saved for retraining $F_\theta$. In addition, the semantic labels corresponding to the visual samples are extracted from the simulator and used as supervisory signals for these two tasks.

**Object Navigation.** The pre-trained $F_\theta$ and $F_\varphi$ are integrated into the framework illustrated in Fig. 3 to validate their contributions to ObjectNav. Following existing works [45, 61, 62], the ObjectNav task is decoupled into an object localization sub-task and a scene exploration sub-task, which are trained using PPO. $F_\theta$ and $F_\varphi$ are used to extract features from consecutive $l+1$ frames of visual perceptions and the corresponding 3D local semantic maps, respectively. We believe the retrained $F_\theta$ and $F_\varphi$ will deliver object-specific semantics and structural cues to the object localization sub-strategy. Moreover, the retrained $F_\varphi$ can provide directional guidance and 3D-level semantic relations for the exploration sub-strategy. In our work, $F_\theta$ and $F_\varphi$ are implemented as the Mask-RCNN [24] with ResNet-50 [25] backbone and the PointNet [43], respectively.

Following existing works [23, 45, 61, 62], an additional egocentric 2D semantic map $M_{2D}^t \in \mathbb{R}^{U \times W \times H}$ is constructed for the exploration sub-strategy and the deterministic local path planning policy. $W$ and $H$ denote the height and width of $M_{2D}^t$, respectively. Each element in $M_{2D}^t$ corresponds to a cell of size $25cm^2$ in the physical world. Each pixel on the egocentric top-down map is labeled with the corresponding semantic category, represented with a one-hot vector with $U = C + 2$ channels where $C$ is the number of object categories, and the extra two channels indicate obstacles and explored regions. An encoder $F_\vartheta$ consisting of fully convolutional networks is employed to extract the scene layout features in $M_{2D}^t$. The scene

layout features and the 3D geometric cues encoded by $F_\varphi$ are fed into the exploration sub-strategy to predict exploration goals $g_{Exp}^t$.

This paper highlights the significant contributions of ECL pre-trained encoders to both scene exploration and object recognition. As shown in Fig. 3, we employed a similar corner-guided exploration sub-strategy as in 3DAware [62]. Please see Section F of the supplementary material for more details. The target goal $g_{target}^t$ is the set of elements in the 2D semantic map with semantic $c_{traget}$. To be specific, once the probability that part of the point clouds in the current 3D local map belongs to the target category $c_{traget}$ is greater than the threshold predicted by our object recognition sub-policy, the agent recognizes the object target. Then, this part of the point clouds is projected onto the 2D semantic map to localize the positions of the object target $g_{target}^t$. The goals $g_{target}^t$ and $g_{Exp}^t$ from two sub-strategies will be consistently updated during Object-Nav. The Fast Marching Method [49] is used to plan the shortest path from the agent's current location to the goal, which is followed by the agent by taking deterministic actions.

## 4 EXPERIMENTS

### 4.1 Experimental Setups

**Object Detection and Instance Segmentation.** We perform experiments on the Habitat simulator [47] with the Gibson [54] dataset that contains photorealistic 3D reconstructions of real-world environments. Following previous works [8, 9], we use 25 train / 5 val scenes from the Gibson tiny split for our experiments where semantic annotations are available. The agent's observation space consists of $640 \times 480$ RGB-D images. The agent's discrete action space consists of Move_Forward (0.25m), Turn_Left (30°), and Turn_Right (30°). Our agent performs about 819K frames of interactive learning in diverse scenarios to optimize the exploration policy $E^2$-CL, the visual encoder $f_\theta$, and the PCL encoder $f_\varphi$. Our ECL is carried out on four NVIDIA 3090 GPUs and takes about 72 hours. The pseudo-code, hyperparameters, and training curves of ECL are listed in Section B of the supplemental material.

For downstream task retraining, our agent actively acts to collect 10K visual frames in each training scene using $E^2$-CL. That is, a total of 250K visual frames are collected. Meanwhile, the corresponding semantic labels are extracted from the Habitat simulator [47] as supervision signals for retraining object detection and instance segmentation models. Following evaluation setups in previous works [8, 9], we use six common indoor object categories for our experiments: chair, couch, bed, toilet, TV, and potted plant. Thus, 250K frames of images contain a total of about 316K objects. We consider two evaluation settings: **(1) Train Split.** 5K images containing 6393 objects are randomly sampled from 25 in-distribution training scenes (200 images per scene). **(2) Test Split.** 5K images containing 5025 objects are randomly sampled from 5 out-of-distribution test scenes (1000 images per scene).

Our method is implemented by using the collected 250K frames to retrain the ResNet50 pre-trained by ECL. To highlight the contributions of our ECL to object-oriented visual representation learning, the following baselines are selected for comparative study: SimCLR [15], CRL [20], OVRL [55], $Ego^2$-MAP$^\dagger$ [28], Pri3D [29], and MIT [57]. Among them, CRL [20] employs an adversarial contrastive loss for embodied visual representation learning based on only

**Table 1: Performance of different methods on object detection and instance segmentation tasks.**

| Method (Venue) | Train Split | | Test Split | |
|---|---|---|---|---|
| | ObjDet | InstSeg | ObjDet | InstSeg |
| *From Scratch* | 62.93 | 52.94 | 15.01 | 13.20 |
| *ImageNet Supervised* | 70.03 | 60.47 | 22.20 | 19.43 |
| SimCLR [15] (ICML 2020) | 66.37 | 54.82 | 20.17 | 18.40 |
| CRL [20] (ICCV 2021) | 68.21 | 56.54 | 22.45 | 19.61 |
| Pri3D [29] (ICCV 2021) | 70.28 | 59.86 | 25.34 | 23.04 |
| OVRL [5, 55] (ICLR 2023) | 67.56 | 54.82 | 22.23 | 20.18 |
| $Ego^2$-MAP$^\dagger$ [28] (ICCV 2023) | 67.29 | 54.97 | 20.72 | 19.89 |
| MIT [57] (ICCV 2023) | 68.30 | 56.88 | 24.19 | 22.61 |
| From ECL (Ours) | **72.32** | **62.11** | **25.89** | **23.81** |

RGB images. $Ego^2$-MAP [28] proposes a multimodal contrastive representation learning method based on visual features and 2D semantic maps. Since we do not have access to the source code of $Ego^2$-MAP, the RGB images and the 2D semantic maps in our method are utilized to replicate the approach as much as possible. The replicated approach is named $Ego^2$-MAP$^\dagger$. To ensure fair comparisons, the RGB images used to pre-train our method are saved for the pre-training of SimCLR [15] and OVRL [55]. Besides the RGB images, additional 2D semantic maps are collected for the pre-training of $Ego^2$-MAP$^\dagger$. Moreover, another two fundamental baselines are set: **(1)** A raw ResNet50 without pre-training is trained in a supervised manner using the collected 250K frames, which is named *From Scratch*. **(2)** The collected 250K frames are used to retrain the ImageNet-supervised pre-trained weights, which is named *ImageNet Supervised*.

We report the bounding box and the mask AP scores for Object Detection (ObjDet) and Instance Segmentation (InstSeg) tasks, respectively. AP is the average precision averaged over multiple Intersection over Union (IoU) (10 IoU thresholds of .50:.05:.95) and is the primary challenge metric in the COCO dataset [37]. IOU is defined to be the intersection over union of the predicted and ground-truth bounding box or the segmentation mask.

**Object Navigation.** We perform experiments on the MP3D [7] and Gibson [54] datasets with the Habitat simulator [47]. For Gibson, we use 25 train / 5 val scenes from the Gibson tiny split. Following existing works [62], we consider 6 goal categories, including chair, couch, potted plant, bed, toilet, and TV. For MP3D, we use the standard split of 61 train / 11 val scenes with the Habitat ObjectNav dataset [47], which consists of 21 goal categories. In the pre-training phase, the visual encoder $F_\theta$ and the PCL encoder $F_\varphi$ are optimized using our ECL method in diverse Gibson and MP3D scenes, respectively. Subsequently, $F_\theta$ and $F_\varphi$ are integrated into the framework shown in Fig. 3 for object navigation. Both our ECL and ObjectNav agents have $640 \times 480$ RGB-D observation spaces for constructing semantic maps. The discrete action space of our ObjectNav agent consists of Move_Forward (0.25m), Turn_Left (30°), Turn_Right (30°), and Stop. Note that, the RGB-D $\{o_t, d_t\}$ and pose $s_t$ readings are noise-free from simulation. The pre-trained 2D semantic model RedNet [31] and the Mask-RCNN [24] trained with COCO dataset [37] are employed for 2D and 3D semantic mapping on MP3D and Gibson datasets, respectively. For each frame, we randomly sample 512 points for PCL-based 3D semantic construction. That is, the number of points in the 3D local semantic map is $Q^t = 512 \times (l + 1)$. During training, we sample actions every 25

**Table 2: ObjectNav results on MP3D (val). † denotes the results we obtained using the official open-source code.**

| Method (Venue) | MP3D (val) | | | |
|---|---|---|---|---|
| | SR (%) ↑ | SPL (%) ↑ | DTS (m) ↓ | Ext. Data |
| DD-PPO [53] (ICLR 2019) | 8.0 | 1.8 | 6.90 | no |
| FBE [56] (First proposed in 1997) | 22.7 | 7.2 | 6.70 | no |
| ANS [10] (CVPR 2020) | 21.2 | 9.4 | 6.30 | no |
| SemExp [9] (NeurIPS 2020) | 28.3 | 10.9 | 6.06 | no |
| Red-Rabbit [58] (ICCV 2021) | 34.6 | 7.9 | - | no |
| TreasureHunt [40] (ICCV 2021) | 28.4 | 11.0 | 5.58 | yes |
| Habitat-Web [46] (CVPR 2022) | **35.4** | 10.2 | - | yes |
| L2M [22] (ICLR 2022) | 32.1 | 11.0 | 5.12 | no |
| PONI [45] (CVPR 2022) | 27.8 | 12.0 | 5.60 | no |
| OVRL [55] (ICLR 2023) | 28.6 | 7.4 | - | no |
| $Ego^2$-MAP [28] (ICCV 2023) | 29.0 | 10.6 | 5.17 | yes |
| 3D-Aware† [62] (CVPR 2023) | 33.4 | 13.6 | 5.03 | no |
| ECL-ObjectNav (Ours) | 34.8 | **14.7** | **4.95** | no |

steps and use the PPO [48] for both object localization and scene exploration sub-policies.

To evaluate all methods qualitatively, the following three metrics are reported: **(1) Success Rate (SR)**: percentage of successful episodes, **(2) SPL**: success weighted by path length, which measures the efficiency of the agent over oracle path length. SPL ranges from 0 to 1 and higher SPL indicates better model performance. **(3) DTS**: geodesic distance of agent to the object goal at the end of the episode. In addition, we report which methods use external data (Ext. Data) to enhance ObjectNav. Besides using the *Random Sample* and classical FBE as non-learning baselines, we consider the following mainstream baselines in the ObjectNav task: **(1) End-to-end strategies:** DD-PPO [53], Habitat-Web [46], Red-Rabbit [58], TreasureHunt [40], OVRL [55], and $Ego^2$-MAP [28]. **(2) Modular strategies:** FBE [56], ANS [10], SemExp [9], FSE [59], L2M [22], PONI [45], and 3D-Aware [62].

## 4.2 Object Detection and Instance Segmentation

The quantitative comparative results between our method and several baselines on the object recognition tasks are shown in Table 1. Firstly, as expected, both self-supervised and *Imagenet Supervised* baselines have significant performance gains relative to *From Scratch*. This phenomenon reflects that different types of pre-training can improve the models' performance on both tasks. Secondly, our method and Pri3D outperform *Imagenet Supervised* pre-training models on both tasks. These results reflect the potential and superiority of 2D-3D self-supervised contrastive learning, i.e., the 3D scene priors can enrich the 2D visual features. Thirdly, the advantages of embodied contrastive learning over offline self-supervised learning are revealed by comparing CRL with SimCLR (or OVRL). Notably, CRL, SimCLR, and OVRL are retrained using the same 250K visual frames. The difference is that CRL uses an adversarial contrastive loss for interactive visual pretraining, while SimCLR and OVRL are pre-trained offline using saved data.

Fourthly, the experimental results of $Ego^2$-Map† indicate that contrastive learning between 2D visual features and 2D semantic maps yields no significant performance improvement. In contrast, the exchange of information between 2D visual features and 3D geometric cues facilitates the accuracy of object recognition tasks. In particular, Pri3D improves the AP metrics by 3.91 (Train Split)

**Table 3: ObjectNav results on Gibson (val). † denotes the results we obtained using the official open-source code.**

| Method (Venue) | Gibson (val) | | | |
|---|---|---|---|---|
| | SR (%) ↑ | SPL (%) ↑ | DTS (m) ↓ | Ext. Data |
| DD-PPO [53] (ICLR 2019) | 15.0 | 10.7 | 3.240 | no |
| FBE [56] (First proposed in 1997) | 41.7 | 21.4 | 2.63 | no |
| *Random Sample* | 54.4 | 28.8 | 1.92 | no |
| ANS [10] (CVPR 2020) | 67.1 | 34.9 | 1.66 | no |
| SemExp [9] (NeurIPS 2020) | 65.2 | 33.6 | 1.52 | no |
| PONI [45] (CVPR 2022) | 73.6 | 41.0 | 1.25 | no |
| FSE [59] (ICRA 2023) | 71.5 | 36.0 | 1.35 | no |
| 3D-Aware† [62] (CVPR 2023) | 73.8 | 39.6 | 1.39 | no |
| ECL-ObjectNav (Ours) | **74.6** | **41.7** | **1.27** | no |

and 5.17 (Test Split) absolutely on the object detection task relative to the self-supervised learning baseline (SimCLR). Similarly, Pri3D improves the AP metrics by 5.04 (Train Split) and 4.64 (Test Split) absolutely on the instance segmentation task relative to the self-supervised learning baseline (SimCLR). Finally, our method achieves the best AP metrics on two splits of both two tasks. On the one hand, by comparing our method with offline self-supervised learning baselines (methods other than CRL), the results reveal the necessity of embodied contrastive learning using 2D visual features and 3D geometric structures. On the other hand, by comparing it with existing offline 2D-3D contrastive representation learning baselines (Pri3D and MIT), our results demonstrate the superiority of our behavioral awareness and geometric consistency modeling.

## 4.3 Object Navigation

Our ECL-enhanced ObjectNav policy is evaluated on the MP3D(val) and Gibson(val) datasets. As shown in Table 2, our method improves the SR and SPL metrics by 12.1%~26.8% and 7.5%~12.9% compared to the learning-based (DD-PPO) and classical frontier-based (FBE) baselines, respectively. In the comparative studies with SOTA methods (OVRL, $Ego^2$-MAP, and 3D-Aware†), our policy improves the SR and SPL metrics by 1.4% ~ 6.2% and 1.1% ~ 7.3% compared to the end-to-end and modular approaches, respectively. Moreover, our scheme achieves the best DTS metric on MP3D(val). Notably, Habitat-Web achieved the best SR metric on MP3D (Val) thanks to the use of a large amount of external data. Nevertheless, our scheme competes strongly with the external data enhanced methods (TreasureHunt, Habitat-Web, and $Ego^2$-MAP).

As an end-to-end approach, OVRL uses a knowledge distillation-like self-supervised learning method to pre-train the visual encoder. In contrast, $Ego^2$-MAP further introduces additional contrastive pre-training based on 2D visual features and 2D scene priors. However, OVRL and $Ego^2$-MAP perform less well than our method. The superiority of our approach is partly attributed to modeling BA and GC in an embodied manner, and partly to a modular strategy design based on semantic maps. More significantly, our approach likewise outperforms the modular and semantic map-based Object-Nav schemes (SemExp, L2M, and PONI). In particular, our approach outperforms 3D-Aware†, which also attempts to exploit the 3D scene priors. Such experimental results reflect the contributions of our ECL-based pre-trained point cloud encoder and visual encoder for scenario exploration and object recognition tasks. Moreover, our BA and GC-based embodied pretraining provides new ways for active scene perception and object recognition.

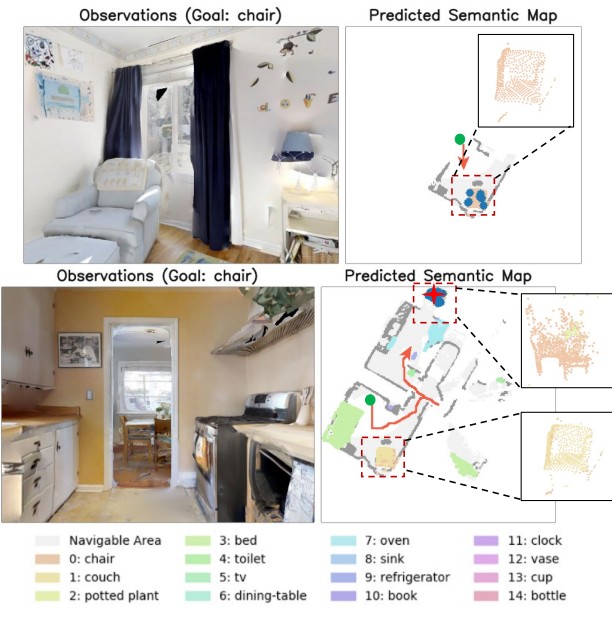

Figure 4: An ObjectNav demo on the Gibson dataset.

As shown in Table 3, our method's performance on Gibson(val) is similar to that on MP3D(val). These experimental results reflect our scheme works universally on different datasets. Quantitatively, our method improves the SR and SPL metrics by 0.8% ∼ 3.1% and 0.7% ∼ 5.7% compared to the SOTA methods (FSE, PONI, and 3D-Aware[†]), respectively. Our method employs an enhanced corner-guided exploration policy, which is distinct from that of SemExp, FSE, and 3D-Aware[†]. This is one of the reasons why we achieve good ObjectNav performance. Qualitatively, a specific ObjectNav example on the Gibson dataset is shown in Fig. 4. The agent initially recognizes the couch as a chair due to a semantic segmentation error. Although the segmentation error has been corrected to some extent, point clouds with the couch semantic in the map are still intermingled with point clouds with chair semantics. Luckily, the features provided by our pre-trained visual encoder motivate the agent to make correct recognition and navigate to a chair in another room. Please see Section H of the supplementary material for more ObjectNav examples.

In addition, we evaluate our method's generalization ability across datasets to verify whether it can handle unseen scenarios, as detailed in Section E of the supplementary material. The experimental results show that ECL pre-training across datasets likewise enhances the ObjectNav performance. As expected, the behavior-aware visual features that fuse 2D-3D cross-modal scene priors can be generalized to novel and unseen scenes.

## 4.4 Ablation Studies

We conduct ablation studies on specific components in the ECL, and the results are shown in Table 4. Specifically, we employ combinations of different components to implement ECL and evaluate the pre-trained models on two object recognition tasks. We first ablate BA $\mathcal{L}_{CE}$ and GC $\mathcal{L}_{ECL}$ while retaining the exploration policy $E^2$-CL ($\mathcal{R}_V$ and $\mathcal{R}_A$) (line 1 ∼ line 3). The results reflect that both BA and GC enhance our method's performance. The best results are

**Table 4: Ablation studies of specific components in ECL.**

| \multicolumn{4}{c}{Ablations} | | | | \multicolumn{2}{c}{Train Split} | | \multicolumn{2}{c}{Test Split} | |
|---|---|---|---|---|---|---|---|
| $\mathcal{R}_V$ | $\mathcal{R}_A$ | $\mathcal{L}_{CE}$ | $\mathcal{L}_{ECL}$ | ObjDet | InstSeg | ObjDet | InstSeg |
| ✓ | ✓ | | | 67.59 | 55.27 | 21.62 | 20.08 |
| ✓ | ✓ | | ✓ | 70.10 | 59.88 | 23.65 | 22.92 |
| ✓ | ✓ | ✓ | | 70.62 | 60.19 | 23.43 | 22.38 |
| ✓ | ✓ | ✓ | ✓ | **72.32** | **62.11** | **25.89** | **23.81** |
| | ✓ | ✓ | ✓ | 67.89 | 56.03 | 21.74 | 20.81 |
| ✓ | | ✓ | ✓ | 69.33 | 58.94 | 24.10 | 21.25 |
| | ✓ | ✓ | ✓ | 70.51 | 59.26 | 25.22 | 23.47 |

**Table 5: Ablation studies of specific features in ObjectNav.**

| \multicolumn{2}{c}{Ablations} | | \multicolumn{3}{c}{Gibson (val)} | | |
|---|---|---|---|---|
| $F_\theta$ | $F_\varphi$ | SR (%) ↑ | SPL (%) ↑ | DTS (m) ↓ |
| ✓ | | 74.4 | 41.5 | 1.28 |
| | ✓ | 74.0 | 41.2 | 1.30 |
| ✓ | ✓ | **74.6** | **41.7** | **1.27** |

achieved when both are used at the same time. In addition, we retain BA and GC and ablate the different components of $E^2$-CL. Line 5 of Table 4 indicates that a randomized wandering exploration method is used. The results show that both $\mathcal{R}_V$ and $\mathcal{R}_A$ enhance our method. Notably, Line 3 of Table 4 shows that better AP metrics can also be achieved by using only $\mathcal{L}_{CE}$ (without $\mathcal{L}_{ECL}$), which suggests that action awareness facilitates the agent's active movement and perception in the scenarios. The comparison with the randomized wandering exploration also demonstrates the superiority of our $E^2$-CL. Please see Section C of the supplementary material for more comparative studies on exploration strategies.

As shown in Table 5, the contributions of $f_\theta$ and $f_\varphi$ are ablated by adopting or not adopting the pre-trained models to initialize the ObjectNav policy. The results show the enhanced ObjectNav performance by using our BA and GC-based 2D-3D multimodal ECL. Both pre-trained $f_\theta$ and $f_\varphi$ can boost the SR and SPL metrics. Additional parametric studies and computational cost evaluation are available in Section D and Section G of the supplemental material, respectively.

## 5 CONCLUSION

This paper focuses on ObjectNav and proposes a novel embodied contrastive representation learning method based on BA and GC, named ECL. In addition, we equip ECL with an exploration strategy $E^2$-CL based on a curiosity and action awareness-driven contrastive reward. Unlike previous self-supervised learning methods based on RGB images, our approach performs an organic cross-modal fusion of semantically rich 2D visual patterns and 3D geometric structural features. In addition, our embodied BA and GC modeling outperforms existing purely vision-based self-supervised learning approaches and offline 2D-3D contrastive representation learning techniques on object recognition tasks. By integrating the pre-trained point cloud encoder and visual encoder into a modular ObjectNav framework, our policy achieves the best performance on the MP3D and Gibson datasets. These improvements are attributed to the adequate scene representation capability and excellent object recognition potential of our pre-trained models. In particular, the ablation studies also indicate the effective contributions of the individual components in our method. In the future, more efficient embodied learning paradigms and 3D feature extraction methods need to be further investigated and discussed.

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
