# OpenReview forum: "Embodied Contrastive Learning with Geometric Consistency and Behavioral Awareness for Object Navigation"
_acmmm.org/ACMMM/2024/Conference — MM2024 Poster_

### Official Review · Reviewer_8t6Z · 2024-05-08

**Rating:** 4
**Confidence:** 4

**Summary:**

This paper introduces the problem of the perception of 2D semantic graphs hindering 3D scene perception and object localization in ObjcetNav.To address this, this paper proposes an Embodied Contrastive Learning (ECL) method with Geometric Consistency (GC) and Behavioral Awareness (BA) is proposed. Furthermore, they propose a curious contrastive reward R𝐸𝑥𝑝 with action awareness, which promotes embodied exploration and encourages agents to gather novel visual images. The authors believe that predicting navigational actions based on multi-frame visual images is crucial for learning correlations among continuous visions. They propose GC to align behavior-aware visual stimuli with 3D semantic shapes through unsupervised contrastive learning. A large number of experimental results prove the effectiveness of the method proposed in the paper.

**Strengths:**

The paper introduces several novel modules:
Geometric Consistency (GC)  aligns 2D observational perception with 3D scene features.
Behavioral Awareness (BA) utilizes continuous visual frames to predict intermediate actions.
A curiosity and action-aware exploration policy 𝐸2 -CL, which continually motivates agents to adopt diverse actions to discover novel visual perceptions.
The approach of aligning 2D observations to 3D features is novel and represents a significant advancement in the field of embodied navigation and perception.
The paper demonstrates thorough experimental work, with results being comprehensive and well-presented.

**Limitations:**

1. Innovation and Differentiation:
While the alignment of 2D and 3D features indeed presents novelty, the modules for action prediction and curiosity-driven exploration are commonly found in reinforcement learning algorithms.
It is essential to emphasize the innovative aspects of the two proposed modules and highlight the distinctions from traditional action prediction and exploration problems in reinforcement learning.

2. Engineering Complexity and Accessibility:
This work leans more towards being an engineering project rather than a research endeavor.
The paper introduces numerous modules, modeling additional information such as semantic map and 3D point cloud data into the visual navigation task. While this engineering effort is commendable, the extensive complexity of the work may limit its accessibility to the research community.

Please elaborate on the contributions of this work to the core research problems in visual navigation, as well as its contributions to the visual navigation community.

**Suitability:**

3

---

### Official Review · Reviewer_KoyP · 2024-05-16

**Rating:** 5
**Confidence:** 3

**Summary:**

The paper addresses limitations in Object Navigation (ObjectNav) by introducing an Embodied Contrastive Learning (ECL) method with Geometric Consistency (GC) and Behavioral Awareness (BA). Current ObjectNav agents struggle with occlusion-prone visual observations or compressed 2D semantic maps, leading to ambiguous object localization and blind exploration. To overcome these challenges, the ECL method motivates agents to actively encode 3D scene layouts and semantic cues. Behavioral Awareness (BA) is modeled by predicting navigational actions based on multi-frame visual images, emphasizing behaviors that cause differences between adjacent visual sensations to learn correlations among continuous visions.

Geometric Consistency (GC) is achieved by aligning behavior-aware visual stimuli with 3D semantic shapes using unsupervised contrastive learning. This alignment, along with geometric invariance priors, enhances the agent's ability to recognize and explore objects. These features are integrated into a modular ObjectNav framework, significantly improving object recognition and exploration capabilities. The ECL method demonstrates superior performance in object detection and instance segmentation tasks and outperforms state-of-the-art methods on the MP3D and Gibson datasets, showcasing its potential in advancing embodied navigation.

**Strengths:**

1. This paper is well-written, and the figures are clear to show the methods;
2. The proposed method shows great improvments and the experiment is well-supported.
3. The method is meaningful, and can be useful for the future paper.

**Limitations:**

The novelty seems a little limited, because contrastive learning method is common used in the object navigation task, I want the authors comment on that.

**Suitability:**

2

---

### Official Review · Reviewer_D7ft · 2024-05-25

**Rating:** 4
**Confidence:** 2

**Summary:**

The authors have studied the problem of object navigation. The goal is to address previous limitations of 3D scene perception. The authors therefore propose an Embodied Contrastive Learning (ECL) method with Geometric Consistency (GC) and Behavioral Awareness (BA). The proposed ObjectNav strategy outperforms previous state-of-the-art methods on MP3D and Gibson datasets. The proposed ECL method also works well on the object detection and instance segmentation tasks.

**Strengths:**

1. The paper is well organised.

2. Extensive experiments and ablation studies have demonstrated the effectiveness of the proposed ECL and GC.

**Limitations:**

Comparison with previous works. There have been lots of works on object navigation. The authors have demonstrated the difference of their work with previous ones in related work. But it would be helpful if there could be a table to summarise the difference more clearly. I may consider to raise my score if my concerns are properly addressed.

**Suitability:**

3

---

### Meta-Review · Area_Chair_o9Qg · 2024-06-30

**Recommendation:** Accept (Poster)
**Confidence:** 5

**Metareview:**

This paper proposes a contrastive learning method to address the issue of the heterogeneous gap between 2D and 3D feature in current embodied navigation methods. In addition, two modules (Geometric Consistency and Behavior Awareness) are proposed to alleviate the mentioned issue. The experiments show the effectiveness in performance compared to previous methods.

All three reviewers agree on the acceptance of this paper. Concerns such as innovation and differentiation are addressed by the author's rebuttal. AC agrees with reviewers that the paper should be accepted. Please revise the paper accordingly based on comments (especially comments from reviewer D7ft).